# Functional Characterization of Splice Variants in the Diagnosis of Albinism

**DOI:** 10.3390/ijms25168657

**Published:** 2024-08-08

**Authors:** Modibo Diallo, Cécile Courdier, Elina Mercier, Angèle Sequeira, Alicia Defay-Stinat, Claudio Plaisant, Shahram Mesdaghi, Daniel Rigden, Sophie Javerzat, Eulalie Lasseaux, Laetitia Bourgeade, Séverine Audebert-Bellanger, Hélène Dollfus, Smail Hadj-Rabia, Fanny Morice-Picard, Manon Philibert, Mohamed Kole Sidibé, Vasily Smirnov, Ousmane Sylla, Vincent Michaud, Benoit Arveiler

**Affiliations:** 1Laboratoire Maladies Rares, Génétique et Métabolisme, Bordeaux University, INSERM U1211, 33076 Bordeaux, France; modibodiallo29@gmail.com (M.D.); cecile.courdier@chu-bordeaux.fr (C.C.); elina.mercier@u-bordeaux.fr (E.M.); angela.sequeira@u-bordeaux.fr (A.S.); alicia.defay-stinat@u-bordeaux.fr (A.D.-S.); sophie.javerzat@u-bordeaux.fr (S.J.); vincent.michaud@chu-bordeaux.fr (V.M.); 2Service de Génétique Médicale, Centre Hospitalier Universitaire de Bordeaux, 33076 Bordeaux, France; claudio.plaisant@chu-bordeaux.fr (C.P.); e.lasseaux@bordeaux.unicancer.fr (E.L.); laetitia.gaston@chu-bordeaux.fr (L.B.); 3Institute of Systems, Molecular and Integrative Biology, University of Liverpool, Liverpool L69 7ZB, UK; shahram.mesdaghi2@liverpool.ac.uk (S.M.); drigden@liverpool.ac.uk (D.R.); 4Computational Biology Facility, MerseyBio, University of Liverpool, Crown Street, Liverpool L69 7ZB, UK; 5Service de Génétique Médicale, Centre Hospitalier Universitaire de Brest, 29200 Brest, France; severine.audebert@chu-brest.fr; 6Service de Génétique Médicale, Centre Hospitalier Universitaire de Strasbourg, 67091 Strasbourg, France; dollfus@unistra.fr; 7Service de Dermatologie, Hôpital Necker-Enfants Malades, 75015 Paris, France; smail.hadj@inserm.fr; 8Service de Dermatologie, Centre Hospitalier Universitaire de Bordeaux, 33076 Bordeaux, France; fanny.morice-picard@chu-bordeaux.fr; 9Hôpital Fondation Rothschild, 75019 Paris, France; mphilibert@for.paris; 10Infirmerie Hôpital Militaire, Bamako BP 236, Mali; iotakole2009@gmail.com (M.K.S.); syllaousmanefr@yahoo.fr (O.S.); 11Service d’Exploration Fonctionnelle de la Vision et de Neuro-Ophtalmologie, Centre Hospitalier Universitaire de Lille, 59037 Lille, France; vasily.smirnov@chu-lille.fr

**Keywords:** albinism, splice variants, exon skipping, pseudoexon, RT-PCR, minigene assay

## Abstract

Albinism is a genetically heterogeneous disease in which 21 genes are known so far. Its inheritance mode is autosomal recessive except for one X-linked form. The molecular analysis of exonic sequences of these genes allows for about a 70% diagnostic rate. About half (15%) of the unsolved cases are heterozygous for one pathogenic or probably pathogenic variant. Assuming that the missing variant may be located in non-coding regions, we performed sequencing for 122 such heterozygous patients of either the whole genome (27 patients) or our NGS panel (95 patients) that includes, in addition to all exons of the 21 genes, the introns and flanking sequences of five genes, *TYR*, *OCA2*, *SLC45A2*, *GPR143* and *HPS1*. Rare variants (MAF < 0.01) in *trans* to the first variant were tested by RT-PCR and/or minigene assay. Of the 14 variants tested, nine caused either exon skipping or the inclusion of a pseudoexon, allowing for the diagnosis of 11 patients. This represents 9.8% (12/122) supplementary diagnosis for formerly unsolved patients and 75% (12/16) of those in whom the candidate variant was in *trans* to the first variant. Of note, one missense variant was demonstrated to cause skipping of the exon in which it is located, thus shedding new light on its pathogenic mechanism. Searching for non-coding variants and testing them for an effect on RNA splicing is warranted in order to increase the diagnostic rate.

## 1. Introduction

Albinism is a clinically and genetically heterogeneous condition characterized by a variable degree of hypopigmentation of the skin, hair and eyes and by ocular features including nystagmus, optic nerve misrouting at the optic chiasma, foveal hypoplasia, retinal hypopigmentation, iris transillumination and reduced visual acuity. Twenty genes are involved in the oculocutaneous (OCA), ocular (OA) and syndromic (Hermansky–Pudlak and Chediak–Higashi Syndromes) forms, and one is involved in the related disease FHONDA (foveal hypoplasia–optic nerve decussation defect–anterior segment dysgenesis syndrome) [1,2]. Analysis of the exonic and directly flanking intronic sequences allows diagnosis for about 70% of patients [3]. For autosomal recessive forms, this relies on the identification of two class 4 (likely pathogenic) or 5 (pathogenic) variants according to the American College of Medical Genetics (ACMG) criteria [4] found in either the homozygous or the compound heterozygous state, each being inherited from one parent.

Among the 30% unsolved patients, about half are found heterozygous for a single pathogenic variant in one of the albinism genes [3]. A second variant *in trans* may be undetected either because it is classified as of uncertain significance (VUS) like most synonymous variants [5,6], or it is located more or less deep in the introns and alters splicing, as it has been observed in various diseases [7,8]. 

The large phenotypical heterogeneity among patients, with incomplete and/or atypical dermatological and ocular presentations, explains that albinism is overlooked at the clinical level. Skin and hair hypopigmentation in particular is highly inconstant, some patients having a normal pigmentation. Ocular symptoms may also be very mild or absent, with so-called micronystagmus, very low grades of iris transillumination, retinal hypopigmentation and foveal hypoplasia, and moderate reduction in visual acuity. This results in an under-diagnosis of this condition [9]. Obtaining the diagnosis is however especially important in order to identify the syndromic forms of the disease that necessitate a specific follow up of the patients due to life-threatening symptoms. In this context, obtaining a precise molecular diagnosis is outstandingly critical. This prompted us to actively search for and characterize the second variant in patients heterozygous for a single class 4 or 5 variant. 

For 122 heterozygous patients, we performed either whole genome sequencing or the sequencing of our albinism next-generation sequencing panel. This panel includes, in addition to all the exons and intron–exon junctions of the 21 genes, the introns as well as 5′ and 3′ gene-flanking sequences for five genes, *TYR* (OCA1), *OCA2* (OCA2), *SLC45A2* (OCA4), *GPR143* (OA1), and *HPS1* (HPS1) (these five genes account for about 90% of cases [3]).

We describe here the identification of exonic and deep intronic variants for which various algorithms predicted an effect on RNA splicing and which were tested functionally by either reverse-transcription PCR or a minigene assay. This approach allowed confirming a splicing alteration effect for nine of them and helped us improve the molecular diagnosis of the disease.

## 2. Results

(A)Analysis flowchart

One hundred and twenty-two patients with a heterozygous likely pathogenic or pathogenic variant (class 4 or 5, according to ACMG; [4]) in one of the known albinism genes were included in this study and analyzed by WGS (27 patients) or using our NGS albinism panel (99 patients). 

Variants located in the exons and introns of the albinism genes were selected and first filtered according to their frequency in the general population with a threshold fixed at a maximum of 0.1% in the gnomAD database (https://gnomad.broadinstitute.org/ accessed on 17 June 2024). Various algorithms were used to predict a possible effect of the variants on RNA splicing (see Section 4).

For 37 variants with a predicted splicing effect, parental segregation was performed whenever possible in order to determine whether this variant was in *cis* or in *trans* to the first class 4 or 5 variant identified in the patient. This was the case for 29 variants. Fourteen variants were found to be in *trans* to the first variant and were subsequently functionally tested by RT-PCR and/or minigene assay, depending on whether the gene under consideration was expressed (*OCA2*, *SLC45A2*; [10]) or not in blood cells, and whether blood-derived RNA samples were available or not for the patient.

A flowchart of the analytical process is shown in Figure 1.

Variants identified in the different patients are presented in Appendix A.

(B)Functional studies of splice variants

(1)*OCA2* variants

Patient 1 had a known pathogenic variant NM_000275.3:c.1503+5G>A inherited from his mother. We found a novel variant in intron 23, NM_000275.3:c.2433-22889T>A, inherited from his father, a rare variant in gnomADv3.1.2 (1 HTZ, 0 HMZ), which was not recorded in databases such as the Human Gene Mutation Database (HGMD Professional 2024.1) (https://www.hgmd.cf.ac.uk/ac/search.php accessed on 17 June 2024) and ClinVar (https://www.ncbi.nlm.nih.gov/clinvar accessed on 17 June 2024). This variant was predicted by various algorithms (see Appendix A) to create a new cryptic splice acceptor site (c.2433-22888) and activate a cryptic splice donor site (c.2433-22728), triggering the inclusion of an intron 23-derived 159 bp pseudoexon, spanning coordinates chr15:g.27778359-27778201 (human hg38 reference, likewise throughout the work presented here), or NM_000275.3:c.2433-22887 to c.2433-22729 (Figure 2A).

RT-PCR was performed on blood-derived RNA with primers designed on one hand in exon 21 and the predicted pseudoexon and, on the other hand, in the predicted pseudoexon and exon 24, which allowed for amplifying 360 bp and 230 bp fragments, respectively (Figure 2B,C). Sanger sequencing of the two products revealed the expected pseudoexonic sequences and junctions between the pseudoexon and exons 23 and 24 (Figure 2D). Insertion of the pseudoexon in the aberrant transcript was predicted to introduce a premature termination codon after 16 novel amino acids [NM_000275.3:c.2433-22889T>A; p.(Arg811_Leu812ins*17)] (Appendix A) and was therefore classified as pathogenic, thus allowing to establish the diagnosis in the patient.

Additional fragments seen in the patient and control RT-PCR were not sequenced and were deemed non-specific products.

Patient 2 had the common exon 7 deletion [11,12] inherited from his father and a novel variant in intron 19 NM_000275.3:c.2080-158A>G inherited from his mother. This very rare variant (1 HTZ, 0 HMZ in gnomADv3.1.2) was absent from the Human Genome Mutation Database and ClinVar. It was predicted by various algorithms (SpliceAI-visual and RNAsplicer) (see Appendix A) to activate both a potential cryptic splice donor site (c.2080-162) and a cryptic splice acceptor site (c.2080-286), triggering the inclusion of a 123 bp pseudoexon located in intron 19, spanning coordinates chr15:g.27872207-27872085 (hg38) or NM_000275.3:c.2080-285 to c.2080-163. Blood RNA was not available for this patient. A minigene assay (Figure 3A) was designed by cloning in plasmid pSPL3B [13] 727 bp fragments (chr15:g.27872508-27871782; NM_000275.3:c.2080-586 to c.2139+81) harboring either the wild type or the variant sequence. Of note, the cloned fragments contained exon 20 in order to include about 300–400 bp of DNA on either side of the variant position in our minigene assays. Following transfection in HeLa cells, RT-PCR was performed using primers derived from vector exons SD6 and SA2 (Figure 3B).

A 446 bp fragment comprising the expected 123 bp pseudoexon in addition to exon 20 was amplified from cells transfected with the variant version (Figure 3B,C right panel). Bands obtained with the wild type construct showed the following: the lower 263 bp band corresponded to the empty vector; the middle 323 bp band corresponded to correctly spliced exon 20 (*OCA2* exon 20 flanked by exons SD6 and SA2 of the pSPL3 vector) without a pseudoexon (Figure 3C left panel); the upper 390 bp band could not be cloned and remained undetermined.

The 123 bp pseudoexon is in phase with the open-reading frame and is predicted to include 41 amino acids in the translated OCA2 protein between those encoded by exons 19 and 20 [c.2080-158A>G; p.(Glu693_Ala694ins41] (Appendix A).

Since no experimental structure of OCA2 is available, ColabFold [14] was used to build models of both wild-type and mutant proteins to visualize the structural impact of the 41-amino-acid insertion. The model with the highest confidence score was selected for detailed examination. This model indicates that the insertion affects the helical packing of the C-terminal scaffold and transport domains of the OCA2 protein, as evidenced by lower predicted Local Distance Difference Test (pLDDT) scores in this region compared to the wild-type protein (see Figure 3D for pLDDT mapped model). Changes in the protein sequence through exon insertion or deletion can produce regions of the protein structure that lack or are in conflict with the evolutionary covariance signal used by AlphaFold2 to guide modeling. Such regions, therefore, might not be well captured by the predictive model, leading to reduced confidence scores in those regions. The regions with lower pLDDT scores suggest that the model is less certain about the precise 3D structure of the protein in those areas. This is because the inserted or deleted sequences might introduce structural variability or disrupt existing structural motifs. In this case, the C-terminal amphipathic helix (helices parallel to the membrane plane) becomes displaced (see Figure 3E for a schematic comparison between the wild-type and variant proteins). This displacement could disrupt its role as a fulcrum that cradles the transport domain during state transitions [15]. Although only the C-terminal side transport machinery is disrupted, transport activity could potentially be reduced or inhibited, as both the N-terminal and C-terminal work in tandem to translocate the substrate across the melanosomal membrane.

**Figure 3 ijms-25-08657-f003:**
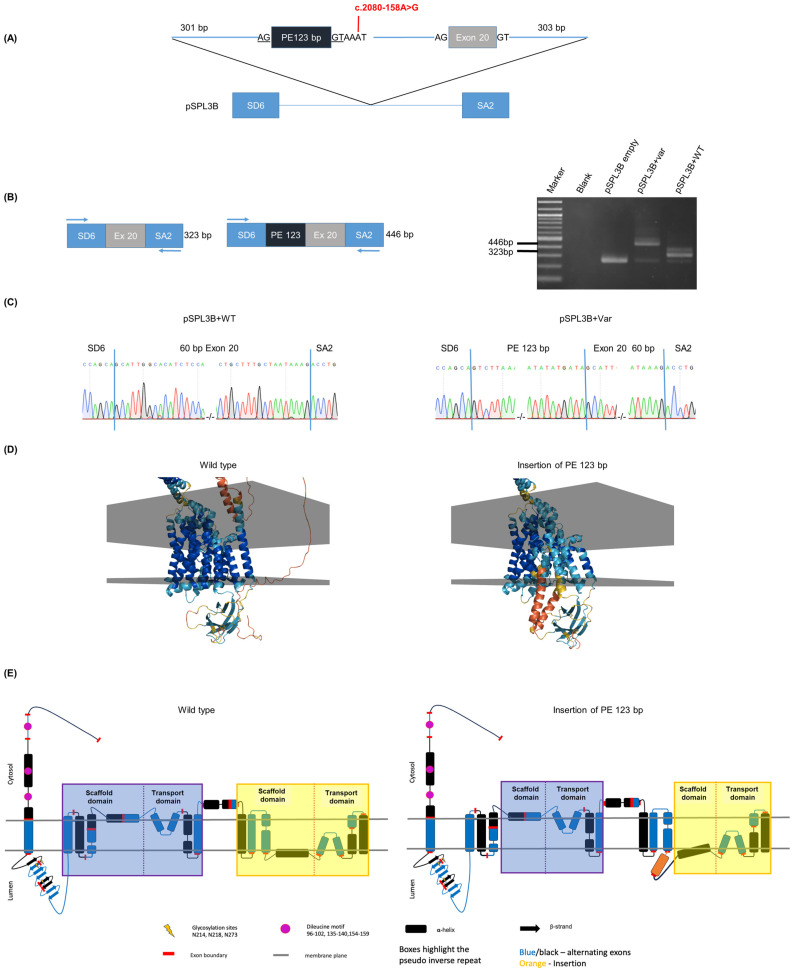
Minigene assay of patient 2 *OCA2* variant c.2080-158A>G. (**A**) Schematic representation of the minigene construct in the vector pSPL3B to test variant c.2080-158A>G. A 727 bp long genomic segment encompassing 646 bp of intron 19 including the variant (in red) and putative pseudoexon (PE 123 bp) as well as exon 20 and 81 bp of intron 20 was cloned in the pSPL3B intron between vector exons SD6 and SA2. Consensus splice site AG and GT dinucleotides at the border of the pseudoexon are underlined. (**B**) Schematic representations of the 323 bp RT-PCR product expected in the absence of inclusion of the pseudoexon and of the 446 bp product expected if the pseudoexon is included. The agarose gel on the right shows the RT-PCR products obtained with the empty vector with the wild-type (WT) insert and the insert carrying the variant (Var). Size marker is a 1 kb ladder. (**C**) Sanger sequencing shows that the WT insert-derived 323 bp RT-PCR product contains only exon 20, and that the variant insert-derived 446 bp RT-PCR product contains exon 20 and the pseudoexon. Vertical blue bars indicate the junctions between exons. (**D**) AlphaFold2 [16] generated OCA2 models. Wild type on the left and variant on the right. Colored by pLDDT (predicted Local Distance Difference Test); blue (high) to red (low)—pLDDT score is a measure of confidence in the predicted structure of a protein. The score ranges from 0 to 100 with higher scores indicating higher confidence in the accuracy of the predicted structure for a particular region of the protein. Gray planes represent membrane boundaries and their positions predicted by the OPM server [17]. (**E**) OCA2 topology: wild type on the left and variant on the right. Wild-type topology adapted from [15] with minor revisions based on new data.

According to ACMG criteria, this variant was classified as likely pathogenic (PM2 PM3 PM4), thus allowing to establish the diagnosis in the patient.

Patient 3 had variants NM_000275.3:c.2339G>A; p.(Gly780Asp) and NM_000275.3:c.1951+1215G>T; p?, which is a very rare variant (2 HTZ, 0 HMZ in gnomADv3.1.2) located in intron 18 of the gene. Bioinformatic predictions using the RNA-Splicer software and SpliceAI-visual (see Appendix A) suggested that this variant could activate a cryptic acceptor splice site (c.1951+1219) and a cryptic donor splice site (c.1951+1296), thereby including an intron 18-derived 77 bp pseudoexon (chr15:g.27950564-27950488) with a premature stop codon in the *OCA2* transcript after 17 amino acids. RT-PCR performed with primers located in exons 16 and 23 showed inclusion of the pseudoexon, as previously described by us [11]. We confirmed this result by a minigene assay in which an 805 bp genomic fragment encompassing the putative pseudoexon was cloned in the pSPL3B vector. The mutated insert led to a 340 bp product that contained the 77 bp pseudoexon (Figure 4A). The wild-type insert assay produced several bands, including the expected empty vector-derived 263 bp RT-PCR product. The 340 bp band containing the 77 bp pseudoexon was also observed, indicating that some leakage may occur, leading to inclusion of this pseudoexon in a proportion of the transcripts, which is a situation that can occur due to the artificial nature of the minigene assay. The upper 400 bp remained of undetermined origin, as its sequence did not blast to any vector or human sequence. Altogether, these functional assays showed that the NM_000275.3:c.1951+1215G>T;p? variant triggered the inclusion of a 77 bp pseudoexon (g.27950564-27950488) in the *OCA2* RNA. Translation of the pseudoexon was predicted to hit a premature stop codon after 17 amino acids, thus producing a truncated P protein [NP_000266.2:p.(Gly651_Trp652ins*18]. This variant was therefore classified as pathogenic (PVS1 PS3 PM2 PM3), allowing to establish the OCA 2 diagnosis in this patient.

Patient 4 had the pathogenic variant NM_000275.3:c.1327G>A; p.(Val443Ile) inherited from his mother and variant NM_000275.3:c.1857C>T; p.(Asp619=) inherited from his father. This rare synonymous variant (247 HTZ, 0 HMZ in gnomAD3.1.2) was predicted by SPiP to impair an exonic splicing regulatory element with 30.67% by the RNAsplicer to decrease both the 5′ splice site (ss) and 3′ ss consensus sites with the same score (0.9956), leading to exon 18 skipping. RT-PCR performed with exon 16- and 23-derived primers did not show the expected band at 616 bp corresponding to exon 18 skipping. Instead, a 725 bp band was observed (Figure 4B), which upon sequencing displayed the presence of exon 18 with the paternal T variant at position 1857 and inclusion of the same intron 18-derived 77 bp pseudoexon as that observed in Patient 3, although patient 4 does not carry the NM_000275.3:c.1951+1215G>T; p? variant. No RNA sample from the patient’s father was available to confirm this finding. This result suggested that the synonymous variant NM_000275.3:c.1857C>T; p.(Asp619=) did not induce exon 18 skipping. However, RNA expressed from the paternal allele included a pathogenic pseudoexon with a premature stop codon in the aberrant transcript, which allowed establishing the diagnosis. Whether or not NM_000275.3:c.1857C>T;p.(Asp619=) was involved in the mechanism leading to inclusion of the pseudoexon will require the analysis of more patients harboring this variant.

Patient 5 had NM_000275.3:c.1117-17T>C inherited from the unaffected father and NM_000275.3:c.1031T>C; p.(Leu344Pro), which is a class 4 likely pathogenic variant (Richard et al., 2015). The mother was not available for segregation analysis. c.1117-17T>C (1601 HTZ, 2 HMZ in gnomADv3.1.2), located in intron 10, was considered a VUS as no splicing effect was predicted by any of the prediction algorithms used (see Appendix A). However, this variant was found in *trans* of the pathogenic or likely pathogenic *OCA2* variant c.1031T>C;p.(Leu344Pro) in several patients addressed for albinism diagnosis to our laboratory (unpublished results). We therefore decided to investigate a potential effect of c.1117-17T>C on *OCA2* RNA splicing. RT-PCR performed on RNA extracted from the patient’s blood cells using primers located in exons 9 and 12 showed three bands at 353 bp, 287 bp and 215 bp (Figure 5A), which were excised from the agarose gel and Sanger sequenced. Sequencing of the 215 bp band showed a mix of three products derived from the paternal allele giving rise to exon 10 and 11 skipping (Figure 5B): one product with normal exon 10 and 11 splicing that was probably not derived from the paternal allele since it carried the c.1031C allele, one derived from the paternal allele with exon 10 and 11 skipping, and one product with exon 10 skipping.

The exclusion of *OCA2* exons 10 and 11 was predicted by the open-source protein-modeling tool Protter (http://wlab.ethz.ch/protter/ accessed on 14 September 2023) to remove three transmembrane domains and to disorganize the remaining domains in the OCA2 protein compared to wild type. Models were also built for this variant using the same methods described for patient 2. The modeling indicated that the deletion of exons 10 and 11 impacts the helical packing of the N-terminal side scaffold and transport domains, which can be seen by the lower pLDDT scores in that region (Figure 5C). Figure 5D visualizes the topological impact of the deletions on the N-terminal scaffold domain—the rigid structure formed by transmembrane and amphipathic helices at the N-terminal scaffold side has been lost. This disruption of the structural integrity of the scaffold domain could severely affect its role to act as a fulcrum that cradles the transport domain as it transitions from state to state, thus causing a reduction or total abolishment of transporter activity.

These results altogether suggested that c.1117-17T>C; p.(Ile349_Met394del) was likely pathogenic (PS3, PM3, BS3 according to ACMG), thus allowing to establish the OCA2 diagnosis in the patient.

(2)*SLC45A2* variants

Patient 6 had in *SLC45A2* a pathogenic variant NM_016180.5:c.264del; p.(Gly89Aspfs*24) inherited from her mother and a novel variant NM_016180.5:c.1157-765C>G inherited from her father. This deep intron 5 variant was absent from gnomADv3.1.2 and was not recorded in the Human Gene Mutation Database and ClinVar. It was predicted by RNA Splicer to create a new splice donor site (c.1157-764) that could recruit a cryptic splice acceptor site at position c.1157-1041, thereby defining a 275 bp pseudoexon. Two RT-PCR experiments were performed on blood-derived RNA with primers designed (i) in exon 5 and the predicted pseudoexon and (ii) in the predicted pseudoexon and at the exons 6–7 junction (Figure 4C). 450 bp and 367 bp fragments were amplified, the Sanger sequencing of which confirmed the presence of the expected pseudoexon between exons 5 and 6. Inclusion of the pseudoexon was predicted to create a STOP codon (TGA) immediately after exon 5. This variant NM_016180.5:c.1157-765C>G; p.(Tyr386*) was therefore classified as pathogenic, allowing to establish the diagnosis of OCA 4 in this patient.

(3)*TYRP1* variants

Patient 7 had two variants in *TYRP1*, NM_000550.3:c.415G>A; p.(Glu139Lys) inherited from his mother and NM_000550.3:c.913+2T>G; p? inherited from his father. The rare (54 HTZ, 0 HMZ in gnomADv3.1.2) missense variant NM_000550.3:c.415G>A; p.(Glu139Lys) was predicted by RNA Splicer to create an alternative splicing acceptor leading to aberrant splicing or total skipping of exon 3. SpliceAI predicted the loss of the 3′ consensus acceptor splice site, and SPiP predicted the alteration of an exonic splicing regulatory element with 47.89% probability. NM_000550.3:c.913+2T>G, p? was predicted to trigger exon 4 skipping. Minigene assays were performed in order to analyze the effect of these two variants on splicing (Figure 4D,E). Analysis of NM_000550.3:c.415G>A (Figure 4D) showed various products in the variant vector transfected cells with a major 263 bp band corresponding to exon 3 skipping. Additional fragments comprised one at about 586 bp and two at about 400 and 500 bp in size. We did not manage to sequence these bands. One of the ~400 and ~500 bp bands may correspond to the aberrant splicing of exon 3 predicted by RNA Splicer. The 586 bp product likely corresponded to the normal splicing of exon 3, since the same product obtained with the wild-type vector was shown by Sanger sequencing to include exon 3. The 263 bp band corresponding to exon 3 skipping was also seen with the wild-type vector but with a lesser intensity than with the variant vector. This is unsurprising with the artificial minigene system known to produce unexpected bands (see for illustrative examples [18,19,20]) such as the upper ~750 bp band observed in the same lane. This may alternatively reflect a propensity of exon 3 for skipping in a proportion of transcripts, although this is not documented in Genotype-Tissue Expression (GTEx, https://gtexportal.org/home/gene/TYRP1#gene-transcript-browser-block, accessed on 17 June 2024).

In total, the NM_000550.3:c.415G>A variant gave rise to abnormal splicing products, with a majority corresponding to exon 3 skipping, and some possibly corresponding to other aberrant splicing events. Only a very low proportion of transcripts, if any, may retain exon 3. The aberrant transcript with exon 3 skipping, would lead, if translated, to a frameshift by changing the valine to glycine at position 129 in exon 2 followed by a premature termination codon after 22 novel amino acids NM_000550.3:c.415G>A; p.(Val129Glyfs23*) (Appendix A) and was classified as pathogenic. We therefore considered that the pathogenicity of this variant was caused by its effect on splicing rather than its missense effect on the TYRP1 protein.

The second variant observed in patient 7, NM_000550.3:c.913+2T>G, p?, was already classified as pathogenic according to ACMG criteria. We confirmed this by a minigene assay, which unequivocally demonstrated exon 4 skipping (Figure 4E), since only the 263 bp band was obtained with the variant vector. The aberrant transcript would lead, if translated, to a frameshift followed by a premature stop codon [c.913+2T>G; p.(Glu237Alafs80*)] (Appendix A).

(4)HPS1 variants

Patient 8 was compound heterozygous for a known pathogenic *HPS1* variant NM_000195.5:c.972dupC; p.(Met325Hisfs*128) inherited from his mother and a novel VUS NM_000195.5:c.1599-16T>G inherited from his father. The latter variant was absent from gnomADv3.1.2 and was not predicted by any software to alter splicing. Specific platelet electron microscopy showed an absence of dense granules, which was consistent with Hermansky–Pudlak syndrome.

Since *HPS1* is expressed in blood leukocytes, we studied the c.1599-16T>G variant by RT-PCR on blood-derived RNA. This showed normal splicing of exon 17 (Figure 6A). We decided to further test this variant by a minigene assay, which displayed a single band at 263 bp, thus indicating exon 17 skipping (Figure 6B). Of note, the wild-type construct showed two bands: one consistent with normal splicing of exon 17 and one with exon 17 skipping, thus indicating that this exon may be prone to alternative splicing. Exon 17 skipping causes a reading frameshift followed by a premature stop codon after 74 amino acids in the aberrant transcript [c.1599-16T>G; p.(Ala170Profs75*)] (Appendix A).

In order to confirm exon 17 skipping, we designed a second RT-PCR assay with primers located in exon 10 where the c.972dupC; p.(Met325Hisfs*128) lies and at the junction between exons 18 and 19 (Figure 6C). Gel electrophoresis revealed a unique PCR product at 961 bp also found in the control and the patient’s mother that corresponded to the normal transcript, i.e., including exon 17. Sanger sequencing (i) confirmed the presence of exon 17, and (ii) displayed the c.972dupC variant nucleotide only, thus demonstrating that the c.1599-16T>G variant-derived transcripts, which would not harbor the c.972dupC, were degraded. Altogether, our data demonstrated that c.1599-16T>G; p.(Ala170Profs75*) caused exon 17 skipping, thereafter resulting in the RNA degradation by non-sense-mediated RNA decay.

(5)Variants for which no splicing effect was detected.

Patients 9, 10 and 11 presented rare *OCA2* (patients 9 and 11) and *TYR* (patient 10) intronic variants in *trans* to a first heterozygous variant. In the absence of available blood samples, these were tested by minigene assay. None of them showed an effect of splicing (see Appendix A).

Patient 12 was previously diagnosed with OCA 1 [11]. Since he had recurrent episodes of epistaxis, a double diagnosis for a syndromic form of the disease was suspected. The patient was found to have two rare synonymous *AP3D1* variants, NM_001261826.3:c.867G>T; p.(Ser289=) in exon 10 (3 HTZ, 0 HMZ in GnomAD3.1.2) and NM_001261826.3:c.3486G>A; p.(Val1162=) in exon 31 (87 HTZ, 0 HMZ in gnomAD3.1.2). Both variants were strongly predicted by SPiP to alter an exonic splicing regulatory element (69.33% and 54% probabilities, respectively), potentially causing exon 10 and exon 31 skipping. These variants were tested by RT-PCR on blood-derived RNA, as the gene is ubiquitously expressed. No evidence of exon skipping was observed for these variants (Appendix A). This allowed us to exclude a diagnosis of HPS 10 in this patient.

## 3. Discussion

Coding variants represent the vast majority of disease-causing single nucleotide variations. However, numerous non-coding variants located deep into the genes’ introns or in flanking sequences and altering either RNA splicing or gene expression have been documented as pathogenic. Molecular analysis of patients with albinism allows an ~70% diagnostic rate [3]. About half of the 30% unsolved patients are heterozygous for a likely pathogenic or pathogenic variant in one of the known albinism genes. We therefore investigated the possibility that the missing variant is located in the introns or the flanking regulatory sequences, which remained unexplored so far.

In the present study, we analyzed 122 heterozygous patients with albinism either by whole genome sequencing or by the use of our next-generation sequencing panel that includes the entire sequence of the *TYR*, *OCA2*, *SLC45A2*, *GPR143* and *HPS1* genes (exons, introns, flanking sequences) (see [11]). Variants with a minor allele frequency ≤ 0.001 were selected.

Among those, 37 variants with a probable effect on RNA splicing predicted by various software were identified in 34 patients: 27 patients had 1 variant, 3 had 2 variants and 4 had 3 variants.

For 14 variants, parental segregation analysis showed that the new variant was in *trans* to the pathogenic variant initially found in the patient. These variants were tested by RT-PCR and/or minigene assays to look for an effect on RNA splicing. RT-PCR on blood-derived RNA was performed for genes expressed in that tissue, i.e., *HPS1* and *AP3D1* (https://gtexportal.org/home/ accessed on 17 June 2024), as well as *OCA2* and *SLC45A2*, which are expressed in blood cells at low levels, but these were sufficient to allow RT-PCR studies [10]. Minigene assays constitute a valuable alternative when the gene is not expressed in blood cells (e.g., *TYR*, *TYRP1*) and skin biopsies are not available and in cases where the gene is expressed in blood cells but adequate samples (PAXgene tubes in our experience) are not available. In this study, six variants were investigated by RT-PCR only, six were investigated by minigene assay only, and two were investigated by both RT-PCR and minigene assay.

Nine variants had a proven effect on splicing, four of which were explored by RT-PCR only, three by minigene only, and two by both RT-PCR and minigene. Five variants were in *OCA2* [NM_000275.3:c.1117-17T>C; p.(Ile349_Met394del) NM_000275.3:c.2080-158A>G; p.(Glu693_Ala694ins41), NM_000275.3:c.2433-22889T>A; p.(Leu812Trpfs17*), NM_000275.3:c.1951+1215G>T; p.(Gly651_Trp652ins*18), NM_000275.3:c.1857C>T; p.(Asp619=)], 2 were in *TYRP1* [NM_000550.3:c.415G>A;p.(Glu139Lys), NM_000550.3:c.913+2T>G, p.(Glu237Alafs80*)], 1 was in *SLC45A2* [NM_016180.5:c.1157-765C>G; p.(Tyr386*)], and 1 in *HPS1* (NM_000195.5:c.1599-16T>G,p.(Ala170Profs75*). Notably, AlphaFold 2 models [16] proved instrumental in providing detailed molecular explanations for the functional defects introduced by the splicing differences.

One variant was at a consensus splice site, two were close to exon–intron junctions (<20 bp into an intron), and four were located deep into an intron, thus indicating that whole gene sequencing is instrumental in identifying pathogenic variants in albinism genes.

One variant was a synonymous variant (NM_000275.3:c.1857C>T; p.Asp619=) (patient 5), which did not trigger exon 18 skipping as predicted by various software, but was associated with the inclusion of an intron 18-derived pseudoexon. It is unclear (i) if and how this variant is directly involved in the mechanism leading to inclusion of this pseudo-exon, (ii) if it is in *cis* with another so far undetected variant that causes inclusion of the pseudoexon, or (iii) if the pseudoexon constitutes an alternative splicing event of the *OCA2* RNA. Analysis of more patients harboring this variant will be necessary to clarify this.

One missense variant, NM_000550.3:c.415G>A; p.(Glu139Lys), was predicted by various bioinformatics tools to alter the splicing of *TYRP1* exon 3. A minigene assay demonstrated exon 3 skipping. Examples of missense variants whose pathogenic effect is in fact due to altered RNA splicing have already been published [7].

Minigene assays are performed by essence in a non-physiological context. In particular, the HeLa cells used here do not express the genes under study. We therefore also performed the minigene assay of variant *OCA2* NM_000275.3:c.1951+605G>T in MNT1, which is a commonly used melanocyte cell line. The same result as that in HeLa cells was obtained. It will be interesting, however, to compare results obtained with more variants, located in *OCA2* or other albinism genes, in these or other cell lines.

The functional tests presented here allowed establishing the diagnosis of albinism in eight patients. In addition, some variants were found in other patients among the 122 analyzed in this study. *OCA2* variant NM_000275.3:c.2433-22889T>A was found in the compound heterozygous state with NM_000275.3:c.1327G>A; p.(Val443Ile) in a patient, the diagnosis of whom was therefore established. Of note, NM_000275.3:c.2433-22889T>A was also compound heterozygous in patient 11 with NM_000275.3:c.1951+605G>T, which did not show any splicing effect in the minigene assay (in both HeLa and MNT1 cells) (see Appendix A). It would have been interesting to investigate this variant further by RT-PCR on blood-derived RNA, but unfortunately, an appropriate sample could not be obtained for patient 11. This variant remains a VUS and the patient is still unsolved, but finding the pathogenic NM_000275.3:c.2433-22889T>A encourages searching for the second pathogenic variant in this gene.

Variant NM_000275.3:c.1951+1215G>T was found in *trans* to likely pathogenic variant NM_000275.3:c.2339G>A; p.(Gly780Asp) in the brother of patient 3 and in another unrelated patient, thus allowing to establish the diagnosis. Of note, the three patients originate from Mali, in western Subsaharan Africa.

*OCA2* variant NM_000275.3:c.1117-17T>C, initially identified in patient 5 was found in another patient from this series in *trans* to likely pathogenic variant NM_000275.3:c.1320G>C; p.(Leu440Phe), allowing to establish the diagnosis. Quite strikingly, c.1117-17T>C was also found in four patients from our larger cohort [21], in whom an *OCA2* pathogenic variant is present. In two of those, segregation analysis showed that c.1117-17T>C was in *trans* to the other, thus establishing the diagnosis. For the remaining two patients, the phase remained undetermined because the parents were not available. NM_000275.3:c.1117-17T>C stands out as a recurrent variant in OCA 2 patients.

Altogether, this study allowed us to establish the diagnosis in 12 patients, representing 9.8% (12/122) supplementary diagnosis for formerly unsolved patients. This percentage is higher if one considers only the 34 patients (35.3%; 12/34) in whom variants with an in silico predicted effect on RNA splicing were identified, and it attains 75% if one solely considers the 12/16 patients in whom the candidate variant was found in *trans* to the first pathogenic variant.

The limitations of the work presented here are mainly technical. Blood cells express the *OCA2* (OCA 2) and *SLC45A2* (OCA 4) genes [10] as well as the Hermansky–Pudlak and Chediak–Higashi genes at sufficient levels to allow RT-PCR analyses. This seems not to be the case for the other genes, especially those coding for the melanogenic enzymes TYR, TYRP1 and DCT. It would, however, be useful to evaluate the feasibility of RT-PCR assays for all the albinism genes in the future. For the genes not expressed in blood cells, obtaining melanocytes from skin biopsies is the main alternative, but this may be not accepted, in particular for children. Hair bulbs could also be used and are more accessible. In the cases where RT-PCR cannot be performed, minigene assays must be performed. These are somewhat more challenging, although they involve standard molecular biology techniques commonly performed in research laboratories. Transferring these assays to clinical laboratories constitutes the next step forward and requires installing specific setups in these laboratories. Both the RT-PCR and minigene approaches may lead to false negative results, thus reflecting their biological limitations. RT-PCR assays may indeed not allow identifying the abnormally spliced RNA due to non-sense-mediated RNA decay, as shown in patient 8. Minigene gene assays are by essence artificial since only a short genomic segment encompassing the variant is analyzed and transfected in cells that may not normally express the gene under study, as is the case for melanogenic genes in HeLa cells. Although changing to the MNT1 melanocyte cell line gave the same result in our hands, using this cell line may be valuable in some instances. Switching to retinal pigmented cells may also be relevant, since a cell-specific (retinal vs. skin-derived) splice factor may be involved, thus leading to different results in different cell lines.

This study shows that searching for non-coding variants and testing those, as well as exonic ones (synonymous, missense), for an effect on RNA splicing is warranted in order to increase the diagnostic rate in patients with albinism. The work presented here is novel for patients with albinism, as no such systematic approach to analyze splice variants has been reported before. First of all, this study demonstrates that the sequencing of whole genes, as we did here with our current diagnostic panel that contains the entirety of five major albinism genes (*TYR*, *OCA2*, *SLC45A2*, *GPR143* and *HPS1*), is instrumental in identifying deep intronic splice variants in these genes. We suggest this approach should be widely implemented by clinical laboratories and possibly extended to the other albinism genes, although these are less frequently mutated. Whole genome sequencing is an alternate possibility that allows searching for non-coding variants in all known albinism genes as well as in other candidate genes or in genes involved in differential diagnosis to albinism. Whether the sequencing of a panel including entire genes or of the whole genome should be performed is a choice for each laboratory, which is dictated by the type of sequencing equipment the laboratory has access to, by the data storage capability and by the cost of analysis, all remaining at the moment several orders of magnitude higher for whole-genome sequencing than for extended panel sequencing. Secondly our study demonstrated that the systematic functional analysis of variants (essentially intronic, but also intra-exonic for some of them) allowed to determine those having an effect on RNA splicing and the exact nature of this effect (i.e., exon skipping, inclusion of a pseudoexon). These tests have a cost at several levels (laboratory setup including bacterial and cell culture rooms, equipment, reagents and consumables, human resources), which is however counterbalanced by the increased diagnostic rate. This is especially crucial and critical when it comes to diagnosing a syndromic form of the disease.

It would be useful if other laboratories adopted a similar strategy to that described here in order to evaluate its operability in different contexts.

In conclusion, we recommend that entire genes are analyzed in order to be able to search for deep intronic variants altering RNA splicing. Our choice was to concentrate on rare variants for which segregation analysis showed they are in *trans* to a first class 4 or 5 variant, thus limiting the number of variants to be tested for each patient. It should be noted, however, that searching for deep intronic variants may prove useful also for the ~15% of patients in whom no variant at all was found by sequencing the exons and exon–intron boundaries [3], since it is possible that these patients harbor two non-coding pathogenic variants, or in consanguineous cases, one variant in the homozygous state. It is also worth reminding that not only very rare variants (MAF < 0.001) should be considered in rare diseases, since less rare variants may be involved, too. This is exemplified by the most frequent *CFTR* pathogenic variant NM_000492.3:c.1521_1523del; p.(Phe508del) in cystic fibrosis (MAF~0.015 in Non-Finnish Europeans) and the *GJB2* variant NM_004004.5:c.35del; p.(Gly12ValfsTer2) in non-syndromic sensorineural deafness (MAF~0.01 in Non-Finnish Europeans).

An alternate and more direct way to analyze the effect of variants on RNA splicing is to perform transcriptome analysis by RNA sequencing. This technique enables seeing all RNA isoforms, physiological and aberrant, of all genes of interest at once, at both the qualitative and quantitative levels. This may constitute a particularly valuable perspective for patients in whom no pathogenic variant at all has been identified at the genomic level. This approach, however, as for RT-PCR analysis, requires access to a tissue in which the gene under consideration is expressed at a sufficient level. A limiting factor is also its cost.

Variants falling in genomic elements annotated as possible regulatory elements by ENCODE (https://genome.ucsc.edu/cgi-bin/hgTracks?db=hg38, accessed on 17 June 2024) [22] should be given full consideration, too. Indeed, pathogenic regulatory variants have been identified in several diseases [21,23], including albinism (see [24,25] for *TYR*, and [26] for *SLC45A2*). This opens yet another research avenue.

The search for pathogenic variants should therefore extend to all gene elements (exons, introns, flanking sequences), and functional analysis should be deployed in order to improve the diagnosis rate of patients with albinism.

## 4. Materials and Methods

(1)Patients

Patients were referred for molecular diagnosis of albinism. All patients, or their parents in the case of minors, gave informed consent for genetic analysis. This study was approved by Bordeaux University Hospital (France) (CHUBX 2021/40; ClinicalTrials.gov ID NCT05696912).

(2)Albinism NGS panel sequencing

The 20 known albinism genes and the gene for FHONDA were analyzed (*TYR*, *OCA2*, *TYRP1*, *SLC45A2*, *SLC24A5*, *LRMDA*, *DCT*, *GPR143*, *HPS1-11*, *LYST*, *SLC38A8*). This included for all genes the exons and intron–exon junctions. The introns and flanking sequences were also analyzed for *TYR* (OCA1), *OCA2* (OCA2), *SLC45A2* (OCA4), *GPR143* (OA1), and *HPS1* (HPS1). Coordinates of all sequences included are defined by comparison with the reference sequence GRCh38 and are available upon request.

Library preparation, capture, enrichment and elution were performed according to the manufacturer’s protocol (SureSelect XT HS Custom; Agilent Technologies). Each sample was sequenced in 75 bp paired-end reads on an Illumina NextSeq550Dx sequencer (Illumina). Alignment on the reference sequence (GRCh38) and variant calling (Single Nucleotide Variants and Copy Number Variants) were performed with Alissa Reporter (Agilent Technologies). Annotation and filtering of the variants were carried out with Alissa Interpret (Agilent Technologies). The sequence of the selected variants was visualized using Alamut Visual Plus (Sophia Genetics). A sample quality data check was performed. Sample identification was performed using the EasySeq™ Human WES WGS Sample Tracking Kit (Nijmegen) according to the manufacturer’s protocol. These identification libraries were then sequenced simultaneously with the Albinism NGS Panel custom libraries. A comparison of the identified SNP was performed with a bioinformatic in-house pipeline to confirm data integrity and validity. Details concerning the analytical method, bioinformatics analysis and versions of the tools and database used are available on request. Segregation analysis of the variants in the parents was performed by Sanger sequencing (BDT v3.1 on ABI3500xL Dx, Thermo Fisher Scientific, Bordeaux, France).

Algorithms used for prediction of an effect on RNA splicing were MaxEntScan through the Alamut Visual Plus aggregator (Sophia Genetics), SPIP [27] and SpliceAI-visual [28], both through the MobiDetails aggregator (https://mobidetails.iurc.montp.inserm.fr/MD/, accessed on 10 January 2024) [29], and RNA Splicer (https://rddc.tsinghua-gd.org/, accessed on 10 January 2014).

Minor allele frequency (MAF) was defined using the Genome Aggregation Database (gnomADv3.1.2) (https://gnomad.broadinstitute.org/ accessed on 10 January 2024) with a threshold ≤0.001.

(3)Whole genome sequencing

Whole genome sequencing was performed by Integragen (Evry, France). PCR free Libraries are prepared with the NEBNext^®^ Ultra™ II DNA Library Prep Kit according to the supplier’s recommendations. Key stages of the protocol are specific double-strand gDNA quantification, sonication of 300 ng of high molecular weight gDNA) to obtain approximately 400 bp fragments, and ligation of paired-end adaptor oligonucleotides (xGen^®^ TS-LT Adapter Duplexes from IDT) on repaired, A-tailed fragments, which were then purified for direct sequencing without PCR step. Libraries are sequenced on an Illumina NovaSeqX as Paired-End 150 (2 × 150). Image analysis and base calling is performed using Illumina Real-Time Analysis with default parameters. Sequence reads are mapped to the human genome build (hg38) by using the Burrows–Wheeler Aligner (BWA) tool. Variant calling for the identification of SNVs (Single Nucleotide Variations) and small insertions/deletions (up to 20 bp) is performed via the Broad Institute’s GATK Haplotype Caller GVCF tool.

Ensembl’s VEP (Variant Effect Predictor, release VEP 101) program processes variants for further annotation. This tool annotates variants, determines the effect on relevant transcripts and proteins, and predicts the functional consequences of variants. It takes into account data available in the gnomAD, the 1000 Genomes Project and the Kaviar databases. An in-house database filters out sequencing artifacts.

Bioinformatics algorithms for pathogenicity prediction are DANN, FATHMM, MutationTaster, SIFT and Polyphen [30,31,32,33,34]. The clinical and pathological significance from the ClinVar database https://www.ncbi.nlm.nih.gov/clinvar, accessed on 15 December 2023) is added. The quality score, homozygote/heterozygote status, and count of variant allele reads are reported. RegulomeDB (https://regulomedb.org/regulome-search/ accessed on 15 December 2023) [35] is used to annotate SNPs in known and predicted regulatory elements in the intergenic regions.

The 96-SNP assay Advanta™ Sample ID Genotyping Panel is used to generate a sample-specific genetic fingerprint in order to confirm the identity between the results delivered and the DNA sample received.

Segregation analysis of the variants in the parents was performed by Sanger sequencing (BDT v3.1 on ABI3500xL Dx, Thermo Fisher Scientific).

(4)Primers

Primers were derived using the commonly used software Primer3 version 4.1.0 [36]. This version contains advanced parameters to specify precise constraints for the primers, such as melting temperature (Tm), amplicons length of the amplicons, GC content, and the presence of secondary structures and of primer dimers. This tool allows for increased primers’ specificity and reduced non-specific amplification, and it is routinely used by us [10].

(5)Sanger sequencing

The variants highlighted by whole genome analysis or targeted diagnostic panel analysis were confirmed by Sanger sequencing with a dual objective of validation and identity vigilance, thus guaranteeing the reliability of the sequencing data. Similarly, RT-PCR products and plasmids used in minigene assays were Sanger sequenced to establish their content and integrity. Sanger sequencing was performed using the Big Dye Terminator v3.1 kit on an ABI3500xL Dx (Thermo Fisher Scientific). Sequences analysis was carried out with the SnapGene Viewer tool (https://www.snapgene.com/snapgene-viewer, accessed on 10 January 2024). This commonly used sequence visualization tool makes it possible to simulate molecular manipulations such as enzymatic digestions, ligations, PCRs and site-directed mutagenesis. Sequenced alignments were performed using BLAT (https://genome.ucsc.edu/cgi-bin/hgBlat accessed on 10 January 2024).

(6)RT-PCR analysis

Total RNA was isolated from white blood cells, using the PAXgene Blood RNA kit (Qiagen), as indicated in the manufacturer’s protocol. First, 1 µg of total RNA from each sample was reverse-transcribed into cDNA using a cDNA synthesis Kit (Thermo Fisher Scientific). Reverse Transcription-PCR primers were designed (primer3 version 4.1.0; https://primer3.ut.ee/, accessed on 10 January 2024) based on the MANE transcript of the mRNA under consideration (*HPS1*: NM_000195.5; *OCA2*: NM_000275.3; *SLC45A2*: NM_016180.5; *AP3D1*: NM_001261826.3; *TYR*: NM_000372.5 and *TYRP1*: NM_000550.3). Primers for RT-PCR and PCR conditions are provided in Appendix A. PCR products were separated by 2% agarose gel electrophoresis and Sanger sequenced (Eurofins).

(7)Minigene assay

Genomic DNA was extracted from peripheral blood leukocytes. Both the wild-type and the variant alleles, flanked by 300–400 bp on each side (details are provided underneath for each variant under consideration), were PCR amplified and cloned in vector pSPL3B by homologous recombination [13]. PCR products were prepared using the “In-Fusion HD Cloning kit with Cloning Enhancer Treatment” (Takara Bio) and recombined with a Not I linearized pSPL3B vector [37]. The wild-type and mutant constructs were Sanger sequenced (Eurofins) to check the presence of the corresponding alleles. Implementation of the minigene assay (cell culture, transfection, reverse-transcription PCR) was performed as in [10]. Details for constructions for each variant are given in Appendix A.

(8)Protein modelization

Protein model building was performed using ColabFold v1.1.5 [14] and implementation of AlphaFold2 [16] on an Ubuntu 20.04.6 workstation AMD Ryzen Threadripper 2990WX 32 Core CPU (3.0 GHz) with 64 GB RAM. GPU acceleration was performed by an ASUS TUF GeForce RTX 3080 OC LHR 12GB GDDR6X Ray-Tracing Graphics Card, 8960 Core, 1815 MHz Boost. The ColabFold ‘template’ search was enabled along with the energy minimization refinement flag (‘amber’) and the GPU relax feature (--use-gpu-relax). The OPM server [17] was used to position the OCA2 models into the lipid bilayer.

### ClinVar Submission

Variants with an effect on splicing were deposited to ClinVar under numbers SCV005049509--SCV005049513 and SCV005049514 and SCV005049515.

## Figures and Tables

**Figure 1 ijms-25-08657-f001:**
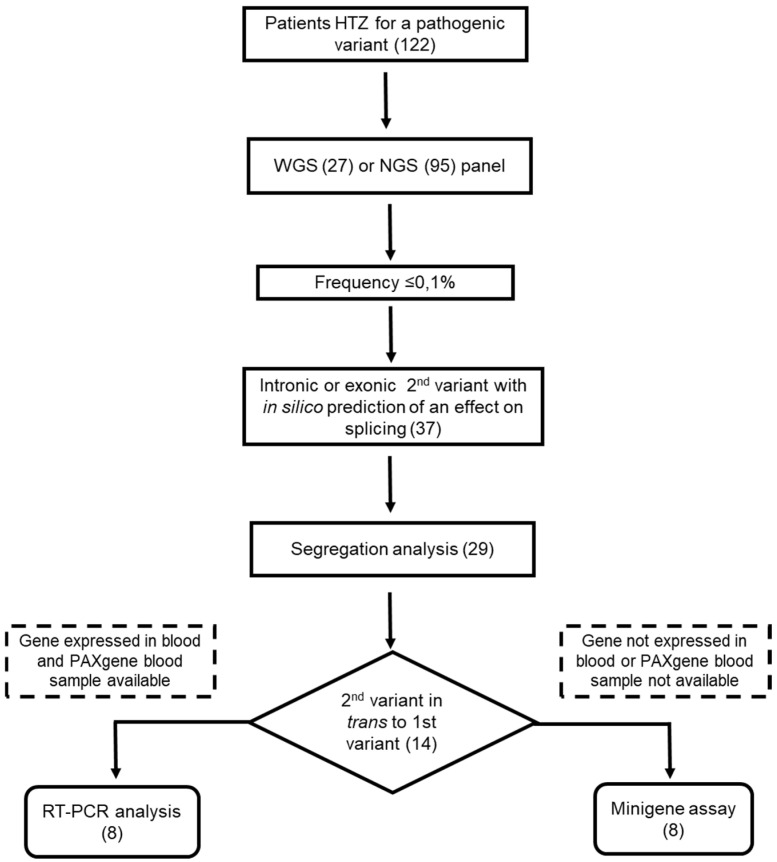
Flowchart of the analytical process. Numbers in parentheses indicate the number of patients, of variants, or of experiments performed.

**Figure 2 ijms-25-08657-f002:**
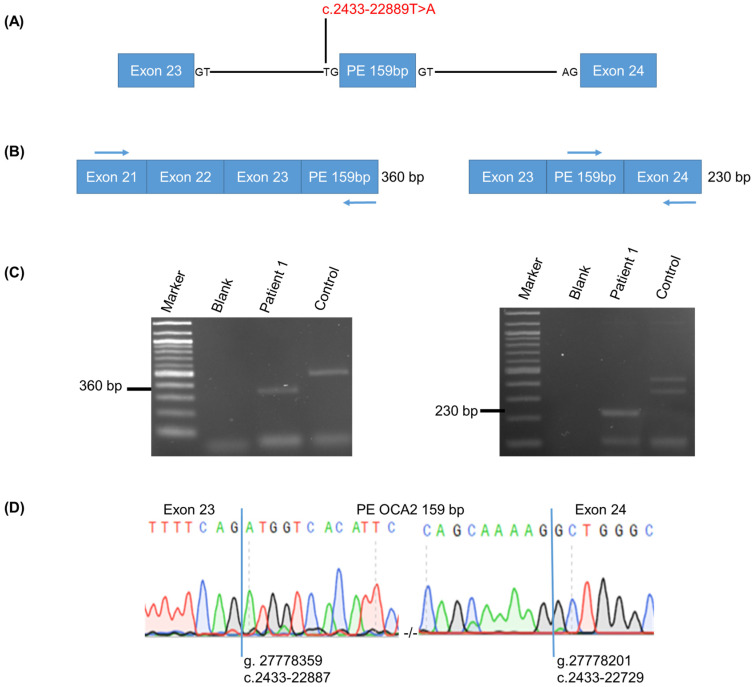
RT-PCR analysis of patient 1 *OCA2* variant c.2433-22889T>A. (**A**) Schematic representation of the genomic region encompassing exon 23 through to exon 24, showing the predicted 159 bp pseudoexon (PE). The c.2433-22889T>A variant is indicated in red. Key intronic nucleotides at consensus splice sites are indicated. (**B**) Schematic representation of the design of the two RT-PCR reactions performed, one extending from exon 21 to the predicted pseudoexon, and one extending from the predicted pseudoexon to exon 24. Primers are indicated by arrows. Sizes of the predicted RT-PCR products are indicated on the right. (**C**) Agarose gels showing the RT-PCR products obtained in the patient and in a control individual without albinism. Sizes of the expected bands in the case of pseudoexon inclusion in the patient are indicated for both RT-PCR reactions. Size marker is a 1 kb ladder. (**D**) Sanger sequences showing the nucleotides at the junction of exon 23 and the 159 bp pseudoexon on the left and at the junction of the 159 bp pseudoexon and exon 24 on the right. Vertical blue bars show where the junctions are.

**Figure 4 ijms-25-08657-f004:**
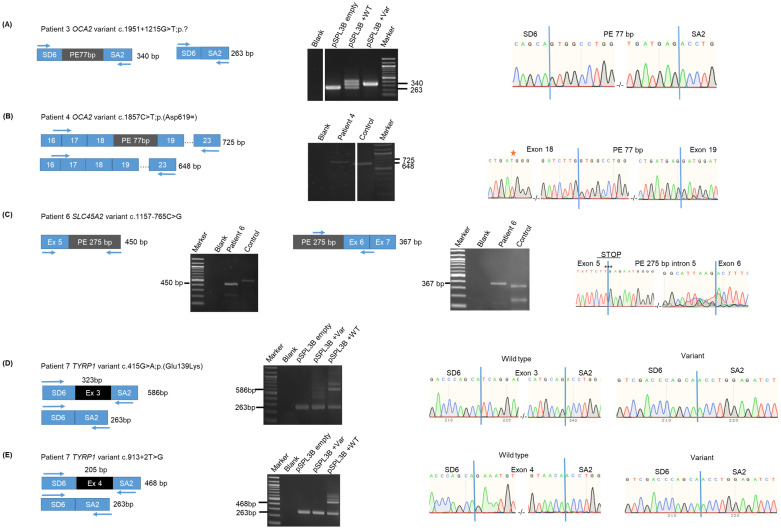
Functional analysis of variants in patients 3, 4, 6 and 7. (**A**) Minigene assay of patient 3 *OCA2* variant c.1951+1215G>T; p? Schematic representations (left) of the 263 bp RT-PCR product expected in the absence of inclusion of the pseudoexon and of the 340 bp product expected if the 77 bp pseudoexon is included. The agarose gel (middle) shows the RT-PCR products obtained with the empty vector with the wild-type (WT) insert and the insert carrying the variant (Var). Size marker is a 1 kb ladder. Sanger sequencing (right) shows that the variant insert-derived 340 bp RT-PCR product contains the 77 bp pseudoexon. Vertical blue bars indicate the junctions between vector-derived exons and the pseudoexon. (**B**) RT-PCR analysis of *OCA2* patient 4 variant c.1857C>T; p.Asp619=. Schematic representations (left) of the 648 bp RT-PCR product in the absence of inclusion of the pseudoexon and of the 725 bp product when the 77 bp pseudoexon is included. Arrows indicate the RT-PCR primers. The agarose gel (middle) shows the 725 bp RT-PCR products in the patient and the 648 bp product in a control individual without albinism. The size marker is a 1 kb ladder. Sanger sequencing (right) shows the inclusion of the 77 bp pseudoexon between exons 18 and 19. Vertical blue bars indicate the junctions between the exons and the pseudoexon. The red star indicates the c.1857C>T variant in exon 18, demonstrating that this sequence corresponds to the variant allele. (**C**) RT-PCR analysis of patient 6 *SLC45A2* variant c.1157-765C>G. Schematic representations and sizes of the expected RT-PCR products resulting from inclusion of the 275 bp pseudoexon using either a exon 5 and a pseudoexon-derived primer or a pseudoexon-derived primer and one located at the junction between exons 6 and 7. Agarose gels display the RT-PCR products obtained for each PCR in the patient and a control individual without albinism. Sizes of the pseudoexon containing bands are indicated in bp. Sanger sequences of part of the exon 5—PE and PE—exon 6/7 RT-PCR products amplified in the patient are shown, focusing on the junctions between exon 5 and PE and PE and exon 6, respectively, thus proving inclusion of the 275 bp PE. The STOP codon created by the inclusion of the pseudoexon (see main text) is indicated. Vertical blue bars indicate the junctions between the exons and the pseudoexon. (**D**,**E**) Minigene assays of patient 7 *TYRP1* variants c.415G>A; p.(Glu139Lys) and c.913+2T>G; p?, respectively. Schematic representations (left) of the expected RT-PCR products corresponding to WT (exon retention) and variant (exon skipping). Sizes of the different products are indicated. Agarose gels (middle) show the RT-PCR products obtained with the empty vector with the wild type (WT) inserts and the insert carrying the variants (Var). Size marker is a 1 kb ladder. Sizes of the observed RT-PCR products are indicated. Sanger sequencing (right) shows exon retention and skipping with the wild type and the variant-carrying constructs, respectively. Vertical blue bars indicate the junctions between vector- and/or insert-derived exons.

**Figure 5 ijms-25-08657-f005:**
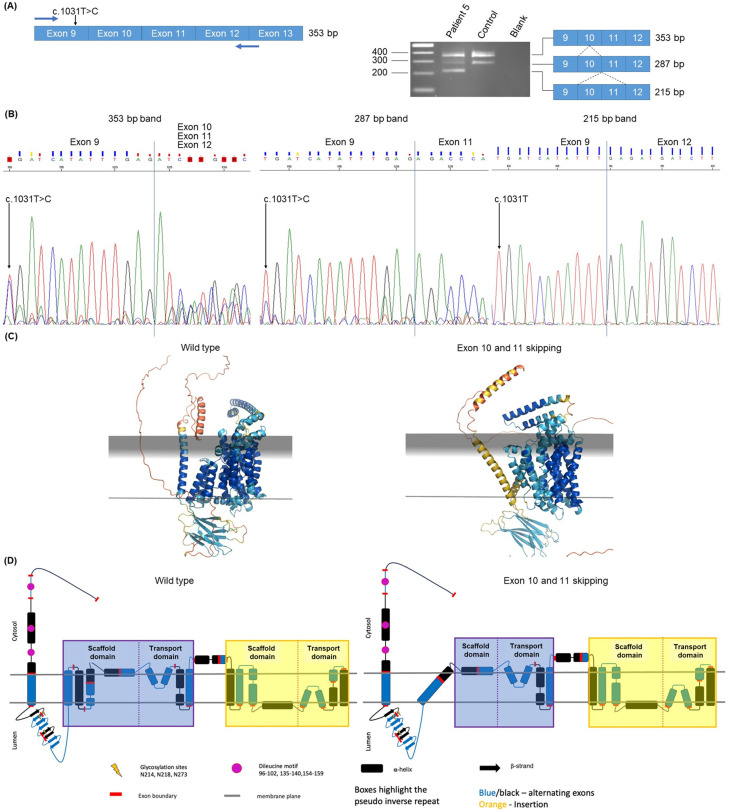
RT-PCR analysis of patient 5 *OCA2* variant c.1117-17T>C. (**A**) Schematic representation of part of the mRNA encompassing exons 9 to 13 (left). Blue arrows represent the RT-PCR primers. The expected correctly spliced 353 bp RT-PCR product is indicated. The c.1031T>C variant (presumably inherited from the mother, see main text) is shown. The agarose gel (right) shows the RT-PCR products obtained in the patient and in a control individual without albinism. The content of the different bands observed is schematized, and sizes are indicated. Size marker is a 1 kb ladder. (**B**) Sanger sequences of part of the 353, 287 and 215 bp bands showing the nucleotides at the junctions between the different exons as indicated for each sequence. Vertical blue bars show where the junctions are. Presence of both C and T or only T at the position of variant c.1031T>C is shown in each sequence. (**C**) AlphaFold2 [16] generated OCA2 models. Wild type on the left and variant on the right. Colored by pLDDT (predicted Local Distance Difference Test); blue (high) to red (low)—pLDDT score is a measure of confidence in the predicted structure of a protein. The score ranges from 0 to 100 with higher scores indicating higher confidence in the accuracy of the predicted structure for a particular region of the protein. Gray planes represent membrane boundaries and their positions predicted by the OPM server [17]. (**D**) OCA2 topology: wild type on the left and variant on the right. Wild-type topology adapted from [15] with minor revisions based on new data.

**Figure 6 ijms-25-08657-f006:**
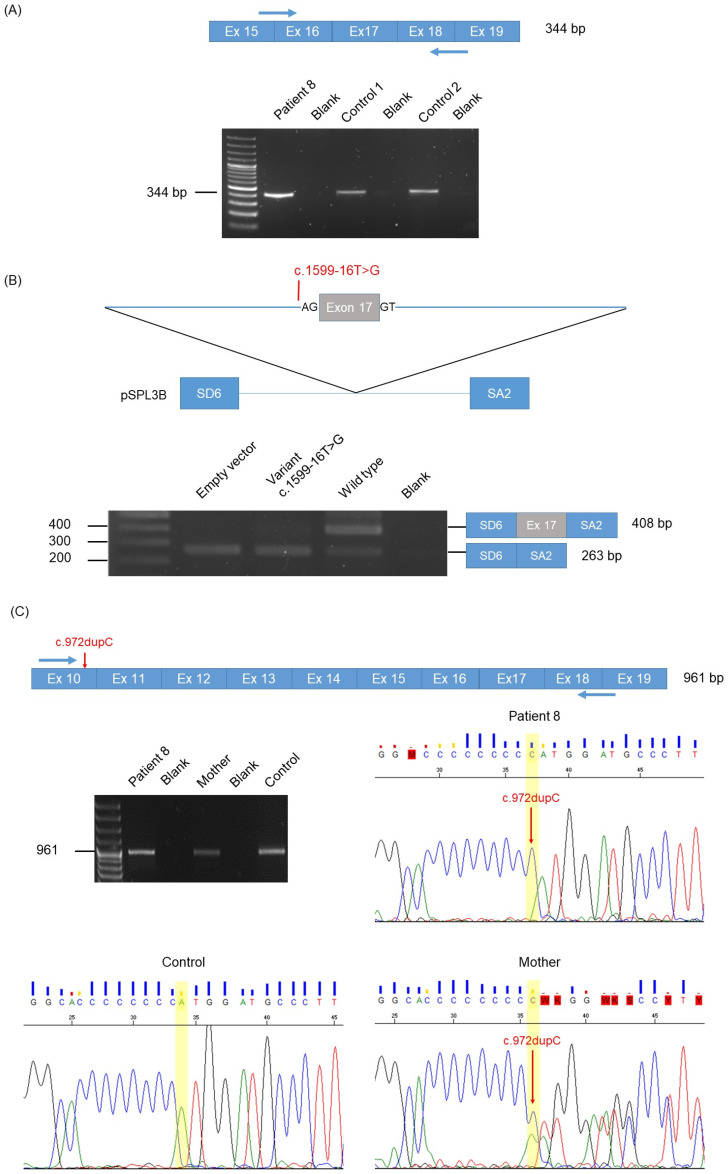
Functional analysis of patient 8 *HPS1* variant c.1599-16T>G. (**A**) RT-PCR analysis between exons 15 and 19. Schematic representation of part of the *HPS1* mRNA encompassing exons 15 to 19. Blue arrows represent the RT-PCR primers. The correctly spliced 344 bp RT-PCR product is indicated. The agarose gel shows the 344 bp product obtained in patient 8 and in 2 control individuals without albinism. Size marker is a 1 kb ladder. (**B**) Minigene assay. Schematic representation of the minigene construct in the vector pSPL3B, displaying *HPS1* exon 17 and variant c.1599-16T>G indicated in red. The agarose gel shows the RT-PCR products obtained with the empty vector with the wild-type insert and the insert carrying the variant. Only the 263 bp product is seen with the variant construct, indicating total exon 17 skipping. The 408 bp product retaining exon 17 is seen in the wild type. Size marker is a 1 kb ladder (bands at 200, 300 and 400 bp are indicated). (**C**) RT-PCR analysis between exons 10 and 19. Schematic representation of part of the *HPS1* mRNA encompassing exons 10 to 19. Blue arrows represent the RT-PCR primers. The correctly spliced 961 bp RT-PCR product is indicated. The maternally inherited c.972dupC variant is in red. The agarose gel shows the 961 bp product in patient 8, his mother and a control individual without albinism. Size marker is a 1 kb ladder. Sanger sequences show that the patient’s RT-PCR product corresponds only to mRNA molecules containing the c.972dupC variant. The mother displays RT-PCR products corresponding to both the c.972dupC variant-derived and wild type-derived mRNA molecules. The control displays only the wild type RT-PCR product, as expected.

## Data Availability

BED files indicating coordinates of the albinism NGS panel used for this study are available from the authors upon request. Variants with an effect on splicing were deposited to ClinVar under numbers SCV005049509--SCV005049513 and SCV005049514 and SCV005049515.

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
