# Peer review of "Functional Characterization of Splice Variants in the Diagnosis of Albinism"

_ijms, 2024, doi:10.3390/ijms25168657_

Round 1

Reviewer 1 Report

Comments and Suggestions for Authors

The number of patients analyzed differs between the abstract/Suppl. Fig 1 (122) and the main text (126). Did I miss the reason why?

Table 1 needs some restyling or should be moved to the supplementary materials. In contrast, the flowchart in supplementary Fig. 1 would be more useful as a main figure.

Regarding patient 1, why was a single PCR spanning exon 23 and 24, including the pseudoexon, not performed? The strange bands present in the control (in both PCRs) should be sequenced to confirm that they are non-specific products. Additionally, the statement that this 159bp pseudoexon causes a frame-shift is incorrect. Since 159 nucleotides correspond to exactly 53 codons, the presence of this aberrant exon should cause an insertion and potentially the introduction of a premature stop codon, not a frame-shift. The mutation name at the protein level should be adjusted accordingly.

For patient 3, it is not clear between which exons the pseudoexon is located. Clarify if this variant has been already described in another paper.

Patient 4 carries a pathogenic missense mutation p.Val443Ile on one allele, which likely doesn’t affect the splicing process. Why didn't an RT-PCR using primers spanning exon 16 to 23 also yield the 648-band from the maternal allele? Since the paternal RNA is unavailable, evaluating this variant using the minigene approach employed for other variants should be considered.

Regarding patient 5, the statement “Sequencing of the 353 bp band showed a mix of three products 280 (Figure 4B): one product with normal exon 10 and 11 splicing that was not derived from the paternal allele since it carried the c.1031C allele” is incorrect. In Fig. 4B, it is impossible to ascertain from which alleles this band is derived, as both nucleotides are present. You can only hypothesize that the wt splicing derived from the maternal allele due to mixed signals. This sentence should be mitigated.

For patient 6, there are also unexpected bands in the control lanes as for patient 1.

Discussion: A valid source of RNA for these genes is hair.

Methods:

Explain Illumina NovaSeqX as Paired End 150 reads, 75x2 or 150x2?

The following sentence is unclear: "Then the libraries were sequenced simultaneously with the SureSelect XT HS custom libraries."

In general, ensure the correct nomenclature of mutations according to www.hgvs.org and improve the style of figure.

Author Response

Comment 1: The number of patients analyzed differs between the abstract/Suppl. Fig 1 (122) and the main text (126). Did I miss the reason why?

Response 1: Thank you for noting this. The exact number is 122. "126" was a mistake that was changed line 84 and line 97.

Comment 2:  Table 1 needs some restyling or should be moved to the supplementary materials. In contrast, the flowchart in supplementary Fig. 1 would be more useful as a main figure.

Response 2: Table 1 was moved to supplementary materials and becomes Supplementary Figure 1. The Flow chart has become a main Figure, ie Figure 1.

Comment 3: Regarding patient 1, why was a single PCR spanning exon 23 and 24, including the pseudoexon, not performed? 

Response 3: This was done, but did not work, as this happens with PCR reactions quite often. Hence we performed these RT-PCRs which target specifically the pseudoexon, thus increasing the sensitivity.

Comment 4: The strange bands present in the control (in both PCRs) should be sequenced to confirm that they are non-specific products.  

Response 4: These bands are present only in the control, not in the patient, and the control does not present the pseudo-exon specific band, which is present in the patient. Sequencing the bands present in the control would be of interest to identify their nature, but this would not change the result that the pseudo-exon is present in the patient only. This kind of bands is usually not sequenced in this field’s literature. For all these reasons, we did not sequence them. We do not have any more RNA to perform the experiment.

Comment 5: Additionally, the statement that this 159bp pseudoexon causes a frame-shift is incorrect. Since 159 nucleotides correspond to exactly 53 codons, the presence of this aberrant exon should cause an insertion and potentially the introduction of a premature stop codon, not a frame-shift. The mutation name at the protein level should be adjusted accordingly.

Response 5: Thank you for this comment. We totally agree with it. We changed the nomenclature to p.(Arg811_Leu812ins*17) (line 148), as well as in Supplementary Table 1 and Supplementary Figure 1.

Comment 6: For patient 3, it is not clear between which exons the pseudoexon is located. Clarify if this variant has been already described in another paper.

Response 6: We agree this needs being clarified, and did so by adding “located in intron 18 of the gene” line 216, “an intron 18-derived” line 218, and “as previously described by us” line 220.

Comment 7A: Patient 4 carries a pathogenic missense mutation p.Val443Ile on one allele, which likely doesn’t affect the splicing process. Why didn't an RT-PCR using primers spanning exon 16 to 23 also yield the 648-band from the maternal allele?

Response 7A: Thank you for your comment. In an initial experiment we performed, with a different set of primers, the band corresponding to the maternal allele was amplified in the patient, but not in the control. This can be seen on the gel attached to the response. We therefore changed primers and cycling conditions. Then the normal band appeared in the control, but not in the patient, despite various attempts.

Comment 7B: Since the paternal RNA is unavailable, evaluating this variant using the minigene approach employed for other variants should be considered.

Response 7B: Since inclusion of the pseudo-exon was observed, we did not consider confirm this by minigene assay, for financial reasons. In addition confirming the presence of the pseudo-exon (PE) by minigene would necessitate a complicated construction ex18-PE-ex19 which would have been tedious to produce without the assurance that it would work successfully.

Comment 8: Regarding patient 5, the statement “Sequencing of the 353 bp band showed a mix of three products 280 (Figure 4B): one product with normal exon 10 and 11 splicing that was not derived from the paternal allele since it carried the c.1031C allele” is incorrect. In Fig. 4B, it is impossible to ascertain from which alleles this band is derived, as both nucleotides are present. You can only hypothesize that the wt splicing derived from the maternal allele due to mixed signals. This sentence should be mitigated.

Response 8: Thank you for this comment. We agree that we only can hypothesis that the wt splicing is derived from the maternal allele. This mixed signal may be due to the retention of shorter products by longer products during electrophoresis. However, we showed that exons 10 and 11 skipping was only observed in the paternal allele (sequencing of the 215 bp band carrying the c.1031T allele). We changed the sentence accordingly (line 280-283):

Sequencing of the 215 bp band showed a product derived from the paternal allele with exon 10 and 11 skipping (Figure 5B). Sequencing of the 353 bp band showed a mix of three products : one product with normal exon 10 and 11 splicing that was probably not derived from the paternal allele since it carried the c.1031C allele, one derived from the paternal allele with exon 10 and 11 skipping, and one product with exon 10 skipping.

Comment 9: For patient 6, there are also unexpected bands in the control lanes as for patient 1.

Response 9: The response to this comment is the same as to Comment 4.

Comment 10: A valid source of RNA for these genes is hair.

Response 10: Thank you for this comment. We added the sentence “Hair bulb could also be used and are more accessible.’ Lines 482 and 483.

Comment 11: Explain Illumina NovaSeqX as Paired End 150 reads, 75x2 or 150x2?

Response 11: We agree this deserves clarification. We added “2x150” line 580.

Comment 12: The following sentence is unclear: "Then the libraries were sequenced simultaneously with the SureSelect XT HS custom libraries."

Response 12: We agree this deserves clarification. We changed the sentence to “These identification librairies were then sequenced simultaneously with the Albinism NGS Panel custom librairies.” Lines 562-563.

Comment 13: In general, ensure the correct nomenclature of mutations according to www.hgvs.org and improve the style of figure. 

Response 13: We carefully checked all the nomenclature of all variants according to HGVS throughout the manuscript, Supplementary Table 1, and the Figures. In addition to correcting p.(Arg811_Leu812ins*17) (already mentioned) we did not identify other big mistakes, but brought several style corrections (adding parentheses when missing, adding or removing spaces where needed). We believe this is fine now.

Concerning the Figures, we had been careful and made efforts to homogenize the Figures’ style in order to help the readers, and facilitate the understanding of their content. We have now brought some modifications in order to further improve them.

Reviewer 2 Report

Comments and Suggestions for Authors

Albinism, while not a common disease among the general population, is a significant condition resulting from genetic defects and holds substantial research value. The authors of this paper present their findings through a compelling and credible narrative, and the characterization of variants for diagnosis, as highlighted in the main theme, is scientifically sound and valid. The paper is detailed and demonstrates considerable value as a scholarly article. However, I have several key suggestions for improving the manuscript for publication.

1. In the introduction section, elaborate on why albinism holds higher research value compared to other genetic disorders, based on the authors' perspectives.

2. Similarly, describe any diseases or symptoms that are commonly associated with the onset of albinism.

3. The PDB code for the protein structure used in Figure 2D should be included in the text or legend, and the related paper should be cited.

4. If not absolutely essential, it is advisable not to submit gel blot data as supplementary material.

5. Provide a brief validation and discussion of the techniques or software used for oligonucleotide analysis within the manuscript.

Minor: The manuscript's formatting appears to be careless. Please pay attention to this during the revision process.

Author Response

Thank you for your positive comments about our work.

Comment 1: In the introduction section, elaborate on why albinism holds higher research value compared to other genetic disorders, based on the authors' perspectives.

Response 1: Many thanks for your suggestion. We have added the following paragraph in the Introduction, lines 75 to 83, which also answer to Comment 2:

“The large phenotypical heterogeneity among patients, with incomplete and/or atypical dermatological and ocular presentations, explains that albinism is overlooked at the clinical level. Skin and hair hypopigmentation in particular is highly inconstant, some patients having a normal pigmentation. Ocular symptoms may also be very mild or absent, with so-called micronystagmus, very low grades of iris transillumination, retinal hypopigmentation and foveal hypo-plasia, and moderate reduction of visual acuity. This results in under-diagnosis of this condition (Arveiler et al., 2020). Obtaining the diagnosis is however especially important in order to identify the syndromic forms of the disease that necessitate a specific follow up of the patients due to life-threatening symptoms. In this context, obtaining a precise molecular diagnosis is outstandingly critical. This prompted us to actively search for and characterize the second variant in patients heterozygous for a single class 4 or 5 variant.”

Comment 2:  Similarly, describe any diseases or symptoms that are commonly associated with the onset of albinism.

Response 2: Thank you for this suggestion. The response is mixed in the paragraph in response to Comment 1.

Comment 3: The PDB code for the protein structure used in Figure 2D should be included in the text or legend, and the related paper should be cited.

Response 3: Thank you for your comment. However, in each figure, the structures are AlphaFold models, not experimental structures. As such there is no PDB code to be mentioned and no paper to be cited. The legends to the Figures do say this. In order to explain this and make it clearer, we changed the previous sentence "ColabFold (Mirdita et al., 2022) was used to build models to visualize the structural impact of the 41-amino-acid insertion." to “Since no experimental structure of OCA2 is available, ColabFold (Mirdita et al., 2022) was used to build models of both wild-type and mutant proteins to visualize the structural impact of the 41-amino-acid insertion.” Line 194-195.  (Of note, Figure 2 has become Figure 3 in this revised version.)

Comment 4: If not absolutely essential, it is advisable not to submit gel blot data as supplementary material.

Response 4: Thank you for your comment. We were not aware of that rule or recommendation. We hence removed all agarose gels from the Supplementary Figures, and changed the legends accordingly. Please see lines 847, 853, 858, 859-862, 868-870.

Comment 5: Provide a brief validation and discussion of the techniques or software used for oligonucleotide analysis within the manuscript.

Response 5: Thank you for your comment. We added 2 chapters to the Materials and Methods section about Primers and Sanger sequencing, lines 598-612, as follows:

4)         Primers

Primers were derived using the commonly used software Primer3 version 4.1.0 (Untergasser et al., 2012). This version contains advanced parameters to specify precise constraints for the primers, such as melting temperature (Tm), amplicons length of the amplicons, GC content, and the presence of secondary structures and of primer dimers. This tool al-lows for increased primers’ specificity and reduced nonspecific amplification, and is routinely used by us (Michaud et al. 2023).

5)         Sanger sequencing

The variants highlighted by whole genome analysis or targeted diagnostic panel analysis were confirmed by Sanger sequencing with a dual objective of validation and identitovigilance, thus guaranteeing the reliability of the sequencing data. Similarly RT-PCR products and plasmids used in minigene assays were Sanger sequenced to establish their con-tent and integrity. Sanger sequencing was performed using the Big Dye Terminator v3.1 kit on an ABI3500xL Dx (Thermo Fisher Scientific). Sequences analysis was carried out with the SnapGene Viewer tool (https://www.snapgene.com/snapgene-viewer). This commonly used sequence visualization tool makes it possible to simulate molecular manipulations such as enzymatic digestions, ligations, PCRs and site-directed mutagenesis. Sequenced alignments were performed using BLAT (https://genome.ucsc.edu/cgi-bin/hgBlat).

Comment 6: The manuscript's formatting appears to be careless. Please pay attention to this during the revision process.

Response 6: Thank you for this comment. We are sorry that you felt we had been careless in the formatting of the manuscript. In order to make the manuscript clearer and more easy to read we introduced numbering of the different parts of the Results and Materials and Methods sections: A), B), 1), 2), 3), …. We also underlined the Patients when we started describing the results for each of them, in order to highlight that a new patient is under discussion.

Reviewer 3 Report

Comments and Suggestions for Authors

Thank you for submitting this interesting outstanding manuscript presenting novel molecular functional tests to establish the diagnosis of albinism.  the manuscript was well written, clear and easy to read.  The topic was  important and would significantly add to our knowledge.  The authors approach was novel and open a new avenues for some pathogenic mechanism that need to be confirmed in future studies.  The conclusion was consistence with the results and supported by enough evidences.  However; I would like to suggest the following to increase the impact of the study:

- Discuss in more details the limitations and challenges of the study and how to combat them in future studies. In additions, please discuss in details the challenges to get it as part the of the clinical practice.

- Highlight the novelty and  the clinical importance and utility of the results of this article. Please explain how adopting this extra tests could help us in the clinical practice.

- the health economic impact for adding this test to the diagnosis path.

- A proposed plan for the next steps to validate the clinical utility of the findings of this article to be widely adopted.

- The future directions and recommendations in more details.

Comments on the Quality of English Language

Minor editing

Author Response

Thank you for your positive comments about our work.

Comment 1: Discuss in more details the limitations and challenges of the study and how to combat them in future studies. In additions, please discuss in details the challenges to get it as part the of the clinical practice.

 Response 1: Many thanks for your comment, which addresses important points in the perspective of clinical practice. We added the following paragraph at lines 477-493:

“Limitations of the work presented here are mainly technical. Blood cells express the OCA2 (OCA 2) and SLC45A2 (OCA 4) genes (Michaud et al., 2023), as well as the Hermansky-Pudlak and Chediak-Higashi genes, at sufficient levels to al-low RT-PCR analyses. This seems not to be the case for the other genes, especially those coding for the melanogenic enzymes TYR, TYRP1 and DCT. It would however be useful to evaluate the feasibility of RT-PCR assays for all the albinism genes in the future. For the genes not expressed in blood cells, obtaining melanocytes from skin biopsies is the main alternative, but this may be not accepted, in particular for children. Hair bulb could also be used and are more accessible. In the cases where RT-PCR cannot be performed, minigene assays must be performed. These are somewhat more challenging although they involve standard molecular biology techniques commonly performed in research la-boratories. Transferring these assays to clinical laboratories constitutes the next step forward and requires installing specific set ups in these laboratories. Both the RT-PCR and minigene approaches may lead to false negative results thus reflecting their biological limitations. RT-PCR assays may indeed not allow identifying the abnormally spliced RNA due to non-sense mediated RNA decay, as shown in patient 8. Minigene gene assays are by essence artificial since only a short genomic segment encompassing the variant is analyzed and transfected in cells that may not normally express the gene under study, as is the case for melanogenic genes in HeLa cells. Although changing to the MNT1 melanocyte cell line gave the same result in our hands, using this cell line may be valuable in some instances. Switching to retinal pigmented cells may also be relevant, since cell specific (retinal vs skin derived) splice factor may be involved, thus leading to different results in different cell lines.”

Comment 2:  Highlight the novelty and  the clinical importance and utility of the results of this article. Please explain how adopting this extra tests could help us in the clinical practice.

Response 2: Thank you again for this comment, which allows us to elaborate on the impact of our study. We added the following paragraph at lines 496-512:

“The work presented here is novel for patients with albinism as no such systematic approach to analyze splice variants has been reported before. First of all this study demonstrates that the sequencing of whole genes, as we did here with our current diagnostic panel that contains the entirety of 5 major albinism genes (TYR, OCA2, SLC45A2, GPR143 and HPS1) is instrumental in identifying deep intronic splice variants in these genes. We suggest this approach should be widely implemented by clinical laboratories and possibly extended to the other albinism genes, although these are less frequently mutated. Whole genome sequencing is an alternate possibility that allows searching for non-coding variants in all known albinism genes, as well as in other candidate genes or in genes involved in differential diagnosis to albinism. Whether sequencing of a panel including entire genes or of the whole genome should be performed, is a choice for each laboratory, that is dictated by the type of sequencing equipment the laboratory has access to, by the data storage capability and by the cost of analysis, all remaining at the moment several orders of magnitude higher for whole genome sequencing than for extended panel sequencing. Secondly our study demonstrated that the systematic functional analysis of variants (essentially intronic, but also intra-exonic for some of them) allowed to determine those having an effect on RNA splicing and the exact nature of this effect (i.e. exon skipping, inclusion of a pseudoexon). These tests have a cost at several levels (laboratory set up including bacterial and cell culture rooms, equipment, reagents and consumables, hu-man resources), which is however counterbalanced by the increased diagnostic rate. This is especially crucial and critical when it comes to diagnose a syndromic form of the disease.”

Comment 3: the health economic impact for adding this test to the diagnosis path.

Response 3: This is an important aspect too. The answer to this comment is included in the paragraph mentioned above (lines 496-512 in the manuscript). Indeed we considered these health economic aspects are intermingled with introducing these tests in clinical practice.

Comment 4: A proposed plan for the next steps to validate the clinical utility of the findings of this article to be widely adopted.

Response 4: In response to this comment we believe it will be important that other laboratories start using the approach we used, in order to validate it in different laboratories. We added the following sentence (lines 513-514).

Comment 5: The future directions and recommendations in more details.

Response 5: Many thanks for this suggestion which allows us to discuss perspectives to this work. We added the following paragraph (lines 515-531):

In conclusion, we recommend that entire genes are analyzed in order to be able to search for deep intronic variants al-tering RNA splicing. Our choice was to concentrate on rare variants for which segregation analysis showed they are in trans to a first class 4 or 5 variant, thus limiting the number of variants to be tested for each patient. It should be noted however that searching for deep intronic variants may prove useful also for the ~15% of patients in whom no variant at all was found by sequencing the exons and exon-intron boundaries (Lasseaux et al., 2018), since it is possible that these patients harbor 2 non-coding pathogenic variants, or in consanguineous cases, 1 variant in the homozygous state. It is also worth reminding that not only very rare variants (MAF < 0.001) should be considered in rare diseases, since less rare variants may be involved too. This is exemplified by the most frequent CFTR pathogenic variant NM_000492.3:c.1521_1523del; p.(Phe508del) in cystic fibrosis (MAF ~0.015 in Non-Finnish Europeans) and the GJB2 variant NM_004004.5:c.35del; p.(Gly12ValfsTer2) in non-syndromic sensorineural deafness (MAF~0.01 in Non-Finnish Europeans).

An alternate and more direct way to analyze the effect of variants on RNA splicing is to perform transcriptome analysis by RNA sequencing. This technique enables seeing all RNA isoforms, physiological and aberrant, of all genes of interest at once, at both the qualitative and quantitative levels. This may constitute a particularly valuable perspective for patients in whom no pathogenic variant at all has been identified at the genomic level. This approach however, as for RT-PCR analysis, requires access to a tissue in which the gene under consideration is expressed at a sufficient level. A limiting factor is also its cost.

Round 2

Reviewer 1 Report

Comments and Suggestions for Authors

The paper changes are satisfactory to me 

Reviewer 3 Report

Comments and Suggestions for Authors

Thank you for addressing my previous comments and I would like to congratulate the authors for this work.